# Higher productivity in forests with mixed mycorrhizal strategies

Shan Luo [1,2,7] ✉, Richard P. Phillips [3,7], Insu Jo [4], Songlin Fei [5], Jingjing Liang [5], Bernhard Schmid [6] & Nico Eisenhauer [1,2]

Decades of theory and empirical studies have demonstrated links between biodiversity and ecosystem functioning, yet the putative processes that underlie these patterns remain elusive. This is especially true for forest ecosystems, where the functional traits of plant species are challenging to quantify. We analyzed 74,563 forest inventory plots that span 35 ecoregions in the contiguous USA and found that in ~77% of the ecoregions mixed mycorrhizal plots were more productive than plots where either arbuscular or ectomycorrhizal fungal-associated tree species were dominant. Moreover, the positive effects of mixing mycorrhizal strategies on forest productivity were more pronounced at low than high tree species richness. We conclude that at low richness different mycorrhizal strategies may allow tree species to partition nutrient uptake and thus can increase community productivity, whereas at high richness other dimensions of functional diversity can enhance resource partitioning and community productivity. Our findings highlight the importance of mixed mycorrhizal strategies, in addition to that of taxonomic diversity in general, for maintaining ecosystem functioning in forests.

Plant diversity can increase ecosystem functioning such as forest productivity[1,2], yet the mechanisms underlying these patterns remain unclear. A primary reason for the uncertainty is that the mechanism frequently invoked to explain positive diversity–productivity relationships—resource complementarity—is notoriously difficult to quantify, particularly in forests. Ecologists have relied on measuring traits of dominant tree species, with the assumption that speciose communities have greater functional trait diversity and hence, greater resource partitioning. However, functional traits can exhibit considerable plasticity within and across species[3], and measuring traits for each species at a site is time-consuming[4]. Thus, trait-based approaches may limit our ability to predict productivity at regional and continental scales, and alternative approaches to understanding how and why plant diversity influences forest functioning are needed.

An important, but understudied, dimension of plant functional variation is plant associations with different mycorrhizal fungal partners[5,6]. Mycorrhizas can shape the spatial distribution of plant diversity[7,8], nutrient cycling[9] and carbon storage[10,11], and have been reported to mediate plant diversity effects on ecosystem functioning in grasslands[12–14]. The role of mycorrhizas in modulating forest productivity is less known, though there are good reasons to believe that a diversity of mycorrhizal associations may be especially important for forest functioning[15,16]. Nearly all tree species associate with one of two types of mycorrhizal fungi: arbuscular mycorrhizal (AM) fungi and ectomycorrhizal (ECM) fungi. The two types of fungi differ substantially in their access to and alteration of nutrient availability, suggesting that stands dominated by trees from both mycorrhizal groups may exhibit complementary (as opposed to competitive) nutrient use[17]. ECM fungi are considered more effective competitors for organic forms of nutrients because of their extracellular enzymatic capabilities[18,19], whereas AM fungi depend on saprotrophic microbes to mineralize nutrients before uptake owing to their limited enzymatic

[1]German Centre for Integrative Biodiversity Research (iDiv) Halle-Jena-Leipzig, Leipzig, Germany. [2]Institute of Biology, Leipzig University, Leipzig, Germany. [3]Department of Biology, Indiana University, Bloomington, IN, USA. [4]Manaaki Whenua – Landcare Research, Lincoln, New Zealand. [5]Department of Forestry and Natural Resources, Purdue University, West Lafayette, IN, USA. [6]Department of Geography, Remote Sensing Laboratories, University of Zürich, Zürich, Switzerland. [7]These authors contributed equally: Shan Luo, Richard P. Phillips. ✉e-mail: luoshan.hi@gmail.com

capabilities[20,21]. ECM fungal mycelium typically proliferates in organic soil horizons, while the AM fungal hyphae occur in upper mineral soil layers, suggesting vertical partitioning of nutrients between fungal groups[22,23]. Moreover, some AM-associating trees exploit nutrient hotspots by proliferating fine roots, in contrast to ECM-associating trees, which use mycorrhizal hyphae[24]. Collectively, the different nutrient acquisition strategies between AM and ECM tree species may also allow them to partition soil nutrients, including different forms of nitrogen and phosphorus[15,25], thereby increasing the total nutrient use of plant communities[15].

In addition to differences in nutrient foraging, AM and ECM tree species possess distinct suites of nutrient-use traits that, in turn, affect nutrient availability. In general, ECM tree species produce low-quality litter that degrades more slowly and inhibits nutrient mineralization, whereas most AM trees produce high-quality litter that degrades more rapidly and promotes nutrient mineralization[9]. In this way, AM and ECM trees represent acquisitive vs. conservative nutrient acquisition strategies[26], and mycorrhizal mixtures—where AM and ECM trees are equally abundant—may exhibit complementary resource use[27]. First principle predicts that productivity should be greatest in mixed mycorrhizal forests where AM and ECM trees are co-dominant[15]. However, there have been few tests of this prediction. This knowledge gap has been a major obstacle for understanding and predicting the functioning of forests under global environmental changes, which can shift relative abundance of AM and ECM tree species[28,29].

In the present study, we examine how tree mycorrhizal dominance (i.e., dominance of trees using a single mycorrhizal strategy vs. mixture of trees using different mycorrhizal strategies) influences forest productivity and whether these processes are related to tree taxonomic diversity across broad spatial scales. We use extensive grid-based forest inventory data from the Forest Inventory and Analysis (FIA) program of USDA Forest Service. The FIA data have been widely used to address ecological questions across environmental gradients[29,30]. Our dataset included 74,563 naturally forested plots (each consisting of four subplots of 168 m²), distributed across the contiguous USA. We used the mean annual increment in tree biomass (total aboveground live biomass divided by stand age) as a measure of forest productivity[1,30] (Supplementary Fig. S1a) and tree species richness as a measure of local (alpha) diversity (Supplementary Fig. S1b). We listed tree species as either AM or ECM[29], excluding tree species with other mycorrhizal strategies that were rare in our dataset (Supplementary Fig. S2). To quantify forest mycorrhizal composition, we calculated AM (or ECM) proportion as the total basal area of AM (or ECM) tree species divided by the total stand basal area. The patterns of ECM and AM proportions are essentially mirror images (Supplementary Fig. S1c, d); therefore, we use AM proportion throughout the text, with increased AM proportion indicating increased dominance of AM tree species.

Two hypotheses guided our research. First, we hypothesized that differences in nutrient use between AM and ECM tree species in a given plot would lead to resource partitioning in mixed mycorrhizal forests, resulting in enhanced forest productivity. Support for this hypothesis would be greater productivity in mixed mycorrhizal stands relative to AM- or ECM-dominated stands[15,25] (i.e., a concave negative relationship between AM tree dominance and forest productivity; Fig. 1a). Second, we hypothesized that the effects of mycorrhizal mixtures on forest productivity would be stronger in species-poor than in species-rich stands, because the latter may make up for a lack of resource partitioning via mycorrhizal associations with other functional-diversity strategies[31]. Alternatively, ECM and AM tree species as groups may overlap more strongly in resource use if there are more species in each group[32–34], again causing weaker effects of mycorrhizal associations in more than in less species-rich stands. Support for this hypothesis would be greater positive effects of mixing mycorrhizal strategies on productivity in species-poor relative to species-rich communities (Fig. 1b, c).

## Results

### Mixing mycorrhizal strategies is related to enhanced forest productivity

We used general linear models to test the effects of AM proportion on forest productivity, with ecoregion, AM proportion (both linear and quadratic terms), tree species richness (log-transformed), interactions between AM proportion and ecoregion, and interactions between AM proportion and richness as explanatory variables. The interaction terms were used to test whether relationships between AM proportion and productivity changed across ecoregions or the species-richness gradient. Additionally, stand age, elevation, slope, climatic variables (i.e., mean annual temperature, mean annual precipitation and temperature seasonality), and soil pH were included as covariates.

We found that ecoregion explained most of the variation in productivity, followed by species richness, which had a significant positive effect on productivity (Model A in Supplementary Table S1: $p < 0.001$ for Ecoregion and Log richness; Supplementary Fig. S3). Nevertheless, linear and quadratic AM proportion still explained a significant amount of variation in productivity, which was more than that explained by environmental covariates (Model A in Supplementary Table S1: $p < 0.001$ for Linear AM proportion and Quadratic AM proportion). As predicted, forest productivity and AM proportion showed a concave-negative relationship across all plots (black line in Fig. 2; see also Supplementary Fig. S4 for a similar pattern with ECM proportion). Moreover, linear and quadratic AM proportion significantly interacted with ecoregion in affecting productivity (Model A in Supplementary Table S1: $p < 0.001$ for AM × ecoregion and AM2 × ecoregion), suggesting that the relationship between forest productivity and AM proportion varied between ecoregions. Thus, 26 out of 34 (~76.5%) ecoregion-level analyses yielded significantly concave-negative relationships between forest productivity and AM proportion (Fig. 2). For the remaining eight ecoregions, four showed significantly linear-negative and the other four showed non-significant relationships between forest productivity and AM proportion (Fig. 2). Overall, these results showed that forests with a mixture of mycorrhizal strategies tended to have higher productivity than forests dominated by either ECM or AM tree species.

To gain a better understanding of the relative importance of AM proportion and other variables in explaining variation in productivity, we fitted random forest models. We included linear and quadratic AM proportion, tree species richness, stand age, climatic variables (i.e., mean annual temperature, mean annual precipitation, and temperature seasonality), soil pH, elevation, slope, and the latitude and longitude of plots in the model. Including latitude and longitude allowed the algorithm to account for spatial variation in productivity. After accounting for spatial variation in productivity, species richness was the most important predictor of productivity, with linear and quadratic AM proportion showing comparable importance (Fig. 3). Mean annual precipitation and soil pH were the most important environmental variables for predicting productivity (Fig. 3).

### Positive effects of mixing mycorrhizal strategies on forest productivity are more pronounced at low than high diversity

In addition, linear and quadratic AM proportion significantly interacted with species richness in affecting productivity (Model A in Supplementary Table S1: $p < 0.001$ for AM × SR and AM2 × SR), which supported our second hypothesis that the relationship between forest productivity and AM proportion would vary with tree species richness. To investigate how tree species richness influenced the relationship between forest productivity and AM proportion, we divided forest plots into two groups depending on tree species richness: low- (five species or less) vs. high-richness (more than five species) plots. We observed a strong concave-negative relationship between AM proportion and productivity in low-richness plots, while there was a weak relationship in high-richness plots (Fig. 4; Supplementary Table S1:

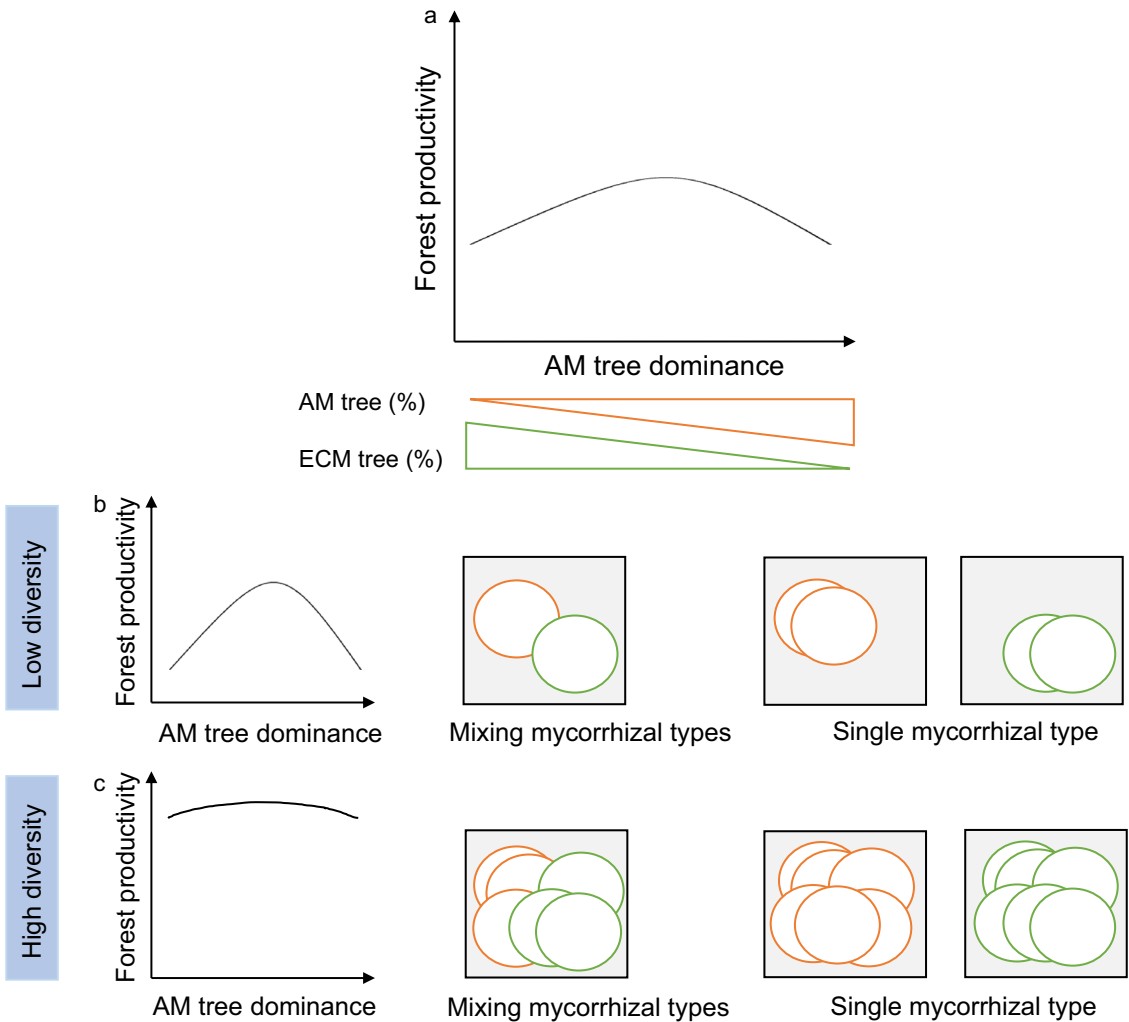

**Fig. 1 | Conceptual figure illustrating the hypothetical relationships between AM tree dominance and forest productivity, as well as the underlying resource-use scenarios in tree species that coexist in local communities. a** Hypothetical overall relationship. **b** Hypothetical relationship at low tree species diversity. **c** Hypothetical relationship at high tree species diversity. We expect a concave-negative relationship between AM tree dominance and forest productivity, which would be more evident at low than at high tree species diversity. At low diversity, we expect that productivity would be higher in communities with both mycorrhizal types than in those dominated by a single mycorrhizal type, because the large niche differences between AM and ECM tree species would maximize the occupied resource space. At high diversity, we expect that resource space would be well occupied by a large number of species, and the positive effects of species diversity on productivity may outweigh that of mycorrhizal composition, which would weaken the relationship between AM tree dominance and productivity. In our illustration, the boxes represent the total available resource space, circles represent the resource space occupied by tree species (orange circles represent AM tree species, green circles represent ECM tree species), and grey areas represent unconsumed resources. We do not illustrate that the resource space occupied by specific species could change with species richness, which merits additional exploration. Rather, we consider simpler cases by assuming that each species has a fixed niche size to abstract some fundamental effects of mixed vs. single mycorrhizal strategies.

$p < 0.001$ for Linear AM proportion and Quadratic AM proportion of Model B; $p < 0.001$ for Linear AM proportion and $p = 0.471$ for Quadratic AM proportion of Model C). In low-richness plots, 23 out of 34 ecoregion-level analyses yielded significantly concave-negative relationships between productivity and AM proportion, while the remaining ecoregions showed significantly linear-negative or non-significant relationships (Supplementary Fig. S5a). However, in high-richness plots only 2 out of 16 ecoregions showed significantly concave-negative relationships (Supplementary Fig. S5b). The stronger relationship between productivity and AM proportion in low- than high-richness plots was robust to different cut-offs of species richness (i.e., four species and six species; Supplementary Fig. S6; Supplementary Table S2).

We further used structural equation models (SEMs) to test how climatic variables (i.e., mean annual temperature, mean annual precipitation, and temperature seasonality) and soil pH modulated the effects of AM proportion (both linear and quadratic terms) on forest productivity in low- and high-richness plots (see Supplementary Fig. S7 for the hypothetical SEM). Consistently, there was a stronger concave-negative relationship between AM proportion and productivity in low- than in high-richness plots (Fig. 5a: std. coef. = 0.18 and std. coef. = –0.39 for AM proportion and AM proportion[2], respectively; Fig. 5b: std. coef. = 0.08 and std. coef. = –0.08 for AM proportion and AM proportion[2], respectively). In low-richness plots, although mean annual precipitation had stronger direct effects on productivity than on AM proportion, mean annual temperature and temperature seasonality had stronger direct effects on AM proportion than on productivity (Fig. 5a). Soil pH had positive and negative direct effects on AM proportion and productivity, respectively. These effects of soil pH were mediated by temperature seasonality. All climatic variables showed direct effects on soil pH. However, the effects of climatic variables and soil pH on

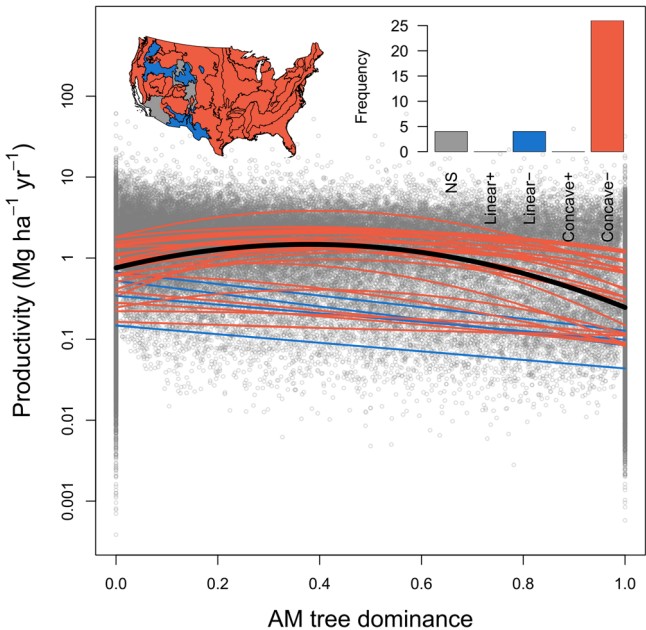

**Fig. 2 | Observed relationship between AM tree dominance and forest productivity.** AM tree dominance is quantified as AM proportion based on tree basal area. The black curve was simple regression fitted across all forest plots, whereas other curves were simple regressions fitted for plots within each ecoregion. Each grey circle represents the data of one forest plot ($n = 74,563$). Inset frequency chart: The frequencies of each form of relationship observed across all ecoregions. Inset map: The colored map indicates the distribution of each form of relationship across ecoregions. The significance of the relationships in the frequency chart and colored map was based on regressions with environmental variables fitted as covariates. NS non-significant. See Supplementary Table S1 for overall statistical results and Supplementary Data 1 and Supplementary Fig. S14 for results and figures of each ecoregion.

AM proportion were weaker in high- than in low-richness plots. In high-richness plots, mean annual precipitation and temperature seasonality had stronger direct effects on productivity than on AM proportion (Fig. 5b). Overall, these results suggested that climatic conditions and soil pH both directly and indirectly influenced productivity by mediating AM proportion, especially in low-richness plots.

## Discussion

Our results show that forests with mixed mycorrhizal strategies achieved higher productivity than forests dominated by a single mycorrhizal strategy both within and across ecoregions in the contiguous USA. The positive effects of mixing mycorrhizal strategies on forest productivity were more pronounced at low than at high tree species richness. Our study indicates that mycorrhizal strategies represent important aspects of plant functional diversity in modulating forest productivity. These findings shed new light on drivers of forest productivity at continental scales, highlighting the role of multiple dimensions of biodiversity on ecosystem functioning.

Consistent with our first hypothesis, forests with mixed mycorrhizal strategies showed higher productivity than forests dominated by a single strategy (Fig. 2). This finding aligns with previous studies showing that the diversity of mycorrhizal fungi can enhance plant community productivity[12,35], potentially by enhancing plant nutrient uptakes via complementary fungal nutrient exploitation strategies[36,37]. Moreover, it has been shown that mycorrhizal fungal diversity was positively related to fine-root trait diversity, which can increase soil resource exploitation efficiency in mixed-species stands[38]. As AM and ECM tree species represent contrasting nutrient acquisition strategies[24,26,37] and can potentially partition soil resources[15,25], we

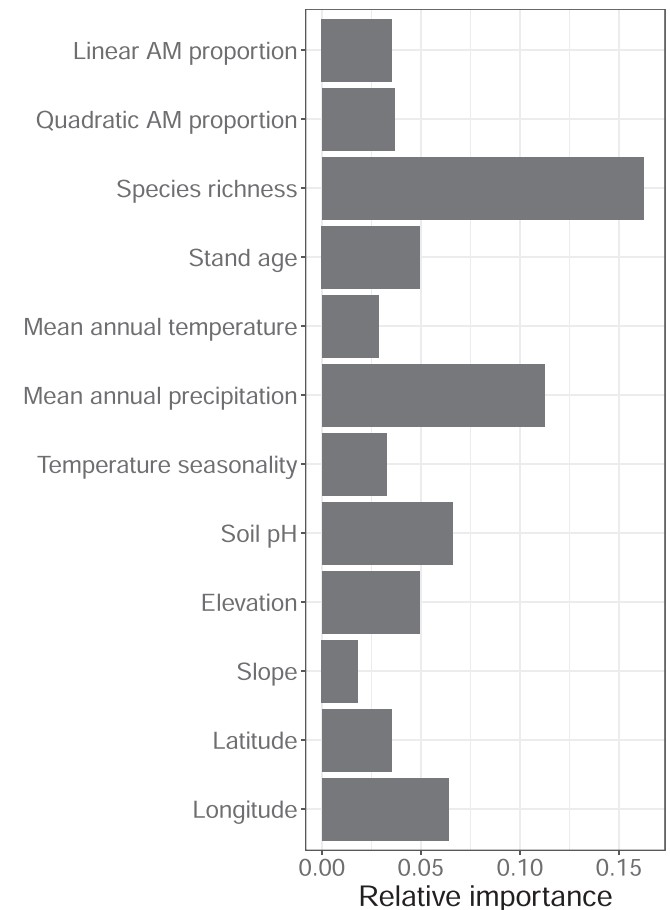

**Fig. 3 | Relative variable importance from random forest model explaining forest productivity.** Relative variable importance is the mean decrease in squared error caused by each of the variables, rescaled such that it sums up to the total pseudo-$R^2$ of the whole model. The overall explained variation ($R^2$) of forest productivity is 0.69. Source data are provided as a Source Data file.

suspect that co-dominant AM and ECM tree species might have increased the overall resource exploitation in stands with mixed mycorrhizal strategies[15,25,39]. This provides additional evidence that variability in plant functional strategies can promote ecosystem functioning[31,40].

In addition to resource partitioning, mixed-mycorrhizal stands may be productive owing to resource enrichment and biotic feedbacks[41]. While ECM litters have been shown to decay more slowly regardless of the soil environment[42,43], herbaceous AM plants in the understory of an ECM-dominated stand can accelerate soil organic matter decomposition[44]. If AM trees facilitate decomposition in a similar way, mixed-mycorrhizal stands may have more nutrients in circulation compared to stands dominated by a single mycorrhizal type. Moreover, it is plausible that mixing mycorrhizal strategies could have non-nutritional benefits, given that AM and ECM tree species generally differ in biotic interactions with other trophic levels[45]. For instance, the mycorrhizal fungal partners may complement each other in protecting host plants against pathogens and herbivores[17]. These differences between AM and ECM tree species may lead to complementarity and enhance productivity in stands with mixed mycorrhizal strategies[41].

As predicted by our second hypothesis, the positive effects of mixing mycorrhizal strategies on productivity were weaker in communities with higher tree species richness (Fig. 4). Multiple coexisting species have been hypothesized to occupy different niche positions[46], whereas the degree of niche overlap between species may increase

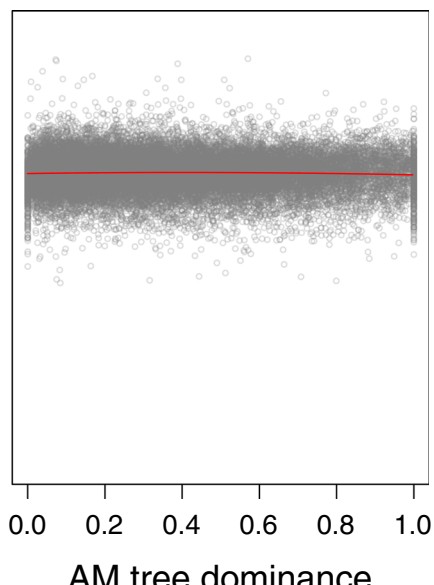

**Fig. 4 | Relationships between AM tree dominance and productivity in forests with low vs. high species richness. a** Forests with low tree species richness (richness ≤ 5). **b** Forests with high tree species richness (richness > 5). We fitted general linear models with ecoregion, AM proportion (linear and quadratic terms), interactions between AM proportion and ecoregion, stand age, elevation, slope, climatic variables, and soil pH as explanatory variables (see Supplementary Table S1 for statistical results).

when communities are saturated with species[47,48]. Particularly, with more species within the groups of ECM and AM tree species, there is greater likelihood that some ECM and AM tree species may overlap in resource use. This may result in weaker effects of mycorrhizal associations in more than in less species-rich stands. However, other types of functional diversity among species may counter this and enhance complementary and facilitative resource extraction in diverse communities[49]. It is thus likely that a large number of tree species can exploit the total resource space via several mechanisms[50] and in this case, the addition of mixed mycorrhizal strategies may have little benefit for increasing resource exploition (Fig. 1c). In contrast, relatively species-poor communities can achieve highest productivity when potential resource partitioning between AM and ECM tree species maximizes the exploited resource space (Fig. 1b). This is consistent with evidence from competition experiments and intercropping showing that resource partitioning is greatest with species differing strongly in functional type, particularly in relatively simple two- or three-species mixtures[51–53].

Why did some ecoregions (23%) not exhibit greater productivity in mixed-mycorrhizal stands? AM tree dominance and forest productivity showed linearly negative correlations in four ecoregions of the western US, where mean annual precipitation was low (Supplementary Fig. S8). The exact mechanisms for explaining the decline of forest productivity are unclear. One possibility is that in arid regions, hydraulic traits of dominant species are more important than nutrient-use traits for productivity, and complementarity among species for water use is less apparent than for nutrient use[54]. Moreover, ECM fungi can transport soil water more efficiently due to greater mycelium biomass and the presence of rhizomorphs[55], which may explain why many ECM tree species are more drought tolerant than co-occurring AM tree species[56,57] and why they tend to dominate in dry climates[58]. Therefore, it is conceivable that the growth of ECM tree species is less limited than that of AM tree species in more water-stressed forests. Additionally, the negative correlation between AM dominance and productivity may relate to the small regional species pool in these arid

regions (Supplementary Table S3). In one eco-region (Black Hills coniferous forest), the ECM-associating *Pinus* comprised 97% of the total basal area (Supplementary Table S4), such that this relatively drought-tolerant species was mostly responsible for the greater productivity in ECM- than AM-dominated stands. Likewise, the dominant AM tree species (i.e., *Prosopis*, *Juniperus*; Supplementary Table S4), though often presumed to be drought tolerant[59,60], often occur as shrubs or small-sized trees with presumably low aboveground productivity. Thus, we suspect that the higher productivity in ECM- than in AM-dominated stands could be attributed to the effects of particular suites of species (i.e., productive ECM tree species or unproductive AM tree species), which could play stronger roles in ecoregions with fewer tree species.

Consistent with other studies showing that climatic conditions and soil pH are strong predictors of dominant mycorrhizal associations[58,61], our results provide support for long-hypothesized environmental controls over plant mycorrhizal strategies[62]. Moreover, climatic conditions and soil pH showed stronger impacts on forest mycorrhizal composition in low- than high-diversity forests. Note that trees with different mycorrhizal associations can also differentially modify soil pH via distinct litter quality and nutrient cycling processes[63,64]. Therefore, a reverse causality from forest mycorrhizal composition on soil pH is likely to co-occur with the direct relationship and disentangling the two is difficult. Nevertheless, our study suggests that global environmental changes may shift forest mycorrhizal composition and consequently influence forest productivity, which can have greater impacts on species-poor than species-rich forests.

While our study presents evidence that mycorrhizal mixtures generally enhance forest productivity, uncertainties remain. First, our study cannot exclude the possibility that particular productive species contribute to the observed relationship between AM tree dominance and productivity (the so-called 'selection effect')[65]. However, the consistent concave-negative relationship between AM tree dominance and productivity in the majority of ecoregions demonstrates that the effect of forest mycorrhizal composition on productivity may be quite

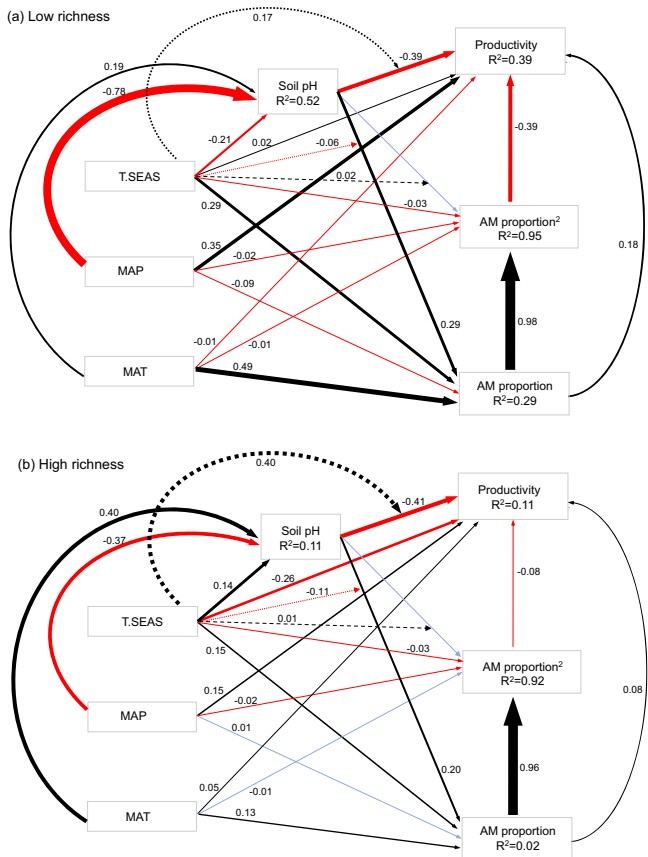

**Fig. 5 | Structural equation models of climate, soil pH, and AM proportion as predictors of forest productivity. a** Forests with low tree species richness (richness ≤ 5). **b** Forests with high tree species richness (richness > 5). Solid black arrows represent positive paths ($p < 0.05$, piecewise SEM), solid red arrows represent negative paths ($p < 0.05$, piecewise SEM), and solid blue arrows represent non-significant paths ($p > 0.05$, piecewise SEM). In addition, we included the interactive effects of T.SEAS and soil pH on AM proportion and productivity, with dashed black and red arrows representing positive and negative effects, respectively. We report the path coefficients as standardized effect sizes. Overall fit of piecewise SEM was evaluated using Shipley's test of d-separation: Fisher's C = 3.466 & $p = 0.177$ for low-richness forests; Fisher's C = 5.206 & $p = 0.074$ for high-richness forests (if $p > 0.05$, then no paths are missing and the model is a good fit). AM proportion², quadratic AM proportion; MAT, mean annual temperature; MAP, mean annual precipitation; T.SEAS, temperature seasonality. Note that the arrows from AM proportion to AM proportion² reflect deterministic relations based on a calculation rather than hypothesized causal relationships.

general. Second, we argue that the ecophysiological differences between mycorrhizal associations might have been overestimated in previous studies, which often, though not always[26], did not correct for phylogenic correlation among tree species. Therefore, caution is needed when using these differences to interpret our results. Third, whether our findings emerge from differences in mycorrhizal traits (between AM and ECM fungi) vs. differences in suites of plant traits (between AM- and ECM-associating tree species) requires further study. Lastly, while our analysis has lumped diverse plants and fungi into AM versus ECM functional groups, we acknowledge that different tree species and mycorrhizal fungi within and across functional groups have diverse ecological strategies[62] and that exploring this diversity may further improve predictions of forest productivity[51]. Despite these knowledge gaps, our approach offers a parsimonious way to capture important aspects of forest community functional variation (relating to resource acquisition and productivity) at large spatial scales. Our approach of classifying forest communities could be especially useful

for predicting ecosystem responses to global change in land surface models[66].

In conclusion, we show that forests with mixed mycorrhizal strategies have higher productivity than those dominated by a single mycorrhizal strategy across the contiguous USA, especially at low levels of tree species richness. Forests with high richness achieve high productivity regardless of tree mycorrhizal strategies, probably via alternative ways of complementary resource uptake between species. Our study indicates the importance of diverse mycorrhizal strategies for maintaining high ecosystem functioning, especially in species-poor forests, where climatic and soil conditions can strongly influence forest mycorrhizal composition. Our findings have important implications for forest management and conservation practices to maintain or improve ecosystem functioning. We suggest that planting tree species with diverse mycorrhizal strategies may be crucial for enhancing productivity in plantations, at least in more mesic regions of temperate forests. Such knowledge is of high relevance, given many major current reforestation/afforestation initiatives, such as in the framework of the UN Decade on Ecosystem Restoration (2021–2030), the Bonn Challenge, and the European Green Deal[67]. In addition, as emerging methods based on remote sensing show promise for generating global estimates of distributions of mycorrhizal associations[68], we suggest that mapping mycorrhizal associations and forest productivity across large spatial and temporal scales would be a next frontier. Given the globally widespread shifts in forest mycorrhizal associations[29,58,69], our study also provides a critical framework for predicting the structure and functioning of forest ecosystems under current and future global-change scenarios.

## Methods
### Forest inventory data
For this study, we used publicly available data from the United States Department of Agriculture's Forest Inventory and Analysis (FIA) program. The FIA program monitors spatiotemporal patterns of forest resources at the continental level, using a fixed grid of permanent plots, which has a sampling intensity of approximately one plot every 2428 ha. Each plot is 0.067 ha (168 m²) and comprises four smaller fixed-radius (7.32 m) subplots spaced 36.6 m apart in a triangular arrangement with one subplot in the center. Diameter at breast height (DBH) are measured and species recorded for all stems with DBH > 12.7 cm. Stand age is measured using dendrochronological records[70]. We subset the original FIA dataset following the protocol of Carteron et al.[8], which only kept census data from the most recent year for a given plot and from natural and undisturbed forests following standardized census methods.

For each plot, we extracted total aboveground biomass, species richness (total number of tree species), stand age, elevation, slope, and physiographic class (estimate of moisture available to trees) from the FIA database. We used mean annual increment in tree biomass (total aboveground live biomass divided by stand age) to estimate forest productivity[1,30], which enabled us to include a considerable amount of plots spreading widely across USA and to test the relatively long-term response of forest growth. Productivity measures based on mean annual increment and periodic annual increment have been shown to be consistent across global forests[1], suggesting that mean annual increment in tree biomass could be a good proxy for productivity in our study. We excluded plots with negative productivity because those are likely to have been disturbed. We used tree species richness to represent local (alpha) diversity. As tree species richness was highly correlated with diversity indices that account for species abundance within the Hill number framework[71], namely the exponential of Shannon's entropy index ($q = 1$; Pearson's correlation = 0.948, $p < 0.001$) and the inverse of Simpson's concentration index ($q = 2$; Pearson's correlation = 0.887, $p < 0.001$), we only presented results based on tree species richness in the main text.

Ecological units are defined as areas of similar surficial geology, lithology, geomorphic processes, soil groups, and sub-regional climate. Following the 'National hierarchical framework of ecological units'[72], we defined 36 ecoregions and assigned each plot to a specific ecoregion depending on its location. We dropped one ecoregion that contains only two plots.

### Tree mycorrhizal strategy and forest mycorrhizal composition

The mycorrhizal strategy for each tree species was assigned based on a recently published database[58], which provides species-level mycorrhizal assignment; 314 out of the 377 species in our study were found in this database. For the remaining 63 species, we extracted genus-level mycorrhizal assignment from another database[73], which assigned genus-level information when > 67% of the observations were consistent. We listed tree species as either AM or ECM, excluding tree species with other mycorrhizal strategies (i.e., non-mycorrhizal or ericoid mycorrhizal) that are rare in our dataset. From the stem diameter measurements, we calculated the total basal area for each species in each plot. We then calculated AM (or ECM) proportion by dividing the total basal area of AM (or ECM) tree species by the total stand basal area in each plot. The basal area of dual AM/ECM tree species was assigned as half AM and half ECM in the calculation of mycorrhizal proportion[29]. Overall, 96.2% of the plots have a cumulative sum of AM and ECM proportion >0.99 (Supplementary Fig. S2), and the patterns of AM and ECM proportions are essentially mirror images (Supplementary Fig. S1c, d). Therefore, we used AM proportion for results in the main text. There was only a weak but significant negative relationship (Supplementary Fig. S9; $R^2 = 0.002$, $p < 0.001$) between AM proportion and stand age.

### Climatic and soil variables

For each selected plot, climatic data were taken from WorldClim (1-km spatial resolution, available at www.worldclim.org/). We used mean annual temperature, mean annual precipitation, and temperature seasonality, which are considered to be predictors of mycorrhizal distribution[58]. We extracted harmonized soil data from GSDE (1-km spatial resolution, available at http://globalchange.bnu.edu.cn/research/soilw) and extracted soil data from FIA database, with the latter including about 3% of all FIA plots. For soil nutrients harmonized data and FIA data are poorly correlated, whereas soil pH from both datasets shows a significant correlation (Person's correlation = 0.60, $p < 0.001$). Therefore, in our analyses we only included soil pH compiled from GSDE, which has been shown to be a good predictor of soil nutrient availability and closely related to tree mycorrhizal dominance patterns[61]. We note that the potential error in pH estimation is likely to underestimate or overestimate the effects of soil pH on productivity. However, this uncertainty applies in the same way to analyses for low- and high-richness plots and thus is unlikely to affect any of our findings or conclusions. We excluded plots with missing values and ended up with 74,563 plots for analyses.

### Statistical analysis

**Effects of forest mycorrhizal composition on productivity.** We tested how forest mycorrhizal composition influenced forest community productivity. First, we used general linear models (GLMs) by fitting forest productivity as the dependent variable, with ecoregion, AM proportion (both linear and quadratic terms), tree species richness (log-transformed), interactions between AM proportion and ecoregion, and interactions between AM proportion and richness as explanatory variables. Stand age, elevation, slope, climatic variables (i.e., mean annual temperature, mean annual precipitation and temperature seasonality), and soil pH were additionally included as covariates. The interaction terms were used to test whether relationships between AM proportion and productivity changed across ecoregions or along the species-richness gradient. Second, we tested how tree species richness modulated the relationships between forest productivity and AM proportion. We divided all forest plots into two groups depending on the number of tree species. We defined plots with five or fewer tree species as low-richness plots and those with more than five species as high-richness plots. We repeated the above GLMs by removing species richness and its interaction term from the model for low- and high-richness plots, respectively. We addressed the potential effects of regional species pool on productivity by fitting ecoregion as the first term in the GLMs, assuming that different ecoregions have different species pools, which can partly account for variation in productivity explained by potential variation in regional species pools. Adding physiographic class in the GLMs hardly reduced the sum of squares of residuals, because it only explained a small fraction of variation in productivity (Supplementary Table S5). Therefore, we excluded physiographic class from all models; this did not influence our main findings and conclusions. Third, we used GLMs to test the relationship between forest productivity and AM proportion in each ecoregion. Again, we included environmental variables (i.e., elevation, slope, mean annual temperature, mean annual precipitation, temperature seasonality, and soil pH) as covariates. For the ecoregion-level analyses, we excluded ecoregions with fewer than 50 plots. All environmental variables were scaled to the range between zero and one to make them comparable to AM proportion, and productivity was log-transformed to ensure normality.

We also tested the relationship between ECM proportion and productivity, which showed a similar pattern as AM proportion (Supplementary Fig. S4). Furthermore, we tested the robustness of the differences in relationships between AM proportion and productivity between low- and high-richness plots by running models with cut-offs of species richness of four and six species (Supplementary Fig. S6; Supplementary Table S2). We additionally tested this robustness by dividing all forest plots into three groups, including low- (six species or less), intermediate- (more than six but less than 13 species), and high-richness (13 species or more) plots (Supplementary Fig. S6; Supplementary Table S2). Using these cut-offs of species richness generated similar patterns as a threshold of five species. Analyses on the effects of tree species diversity on the relationships between forest productivity and AM proportion were conducted for the exponential of Shannon's entropy index ($q = 1$) and the inverse of Simpson's concentration index ($q = 2$), which showed similar patterns as species richness (Supplementary Fig. S10, 11; Supplementary Table S6, 7). In addition, we tested the robustness of our results with extra 2771 managed forest plots that span ten Indiana state forests (USA) from the Continuous Forest Inventory (CFI) project of the Indiana Department of Natural Resources, Division of Forestry[74]. These plots are resampled every five years, which allowed us to measure forest productivity (defined as periodic annual increment in basal area) over shorter time windows (2012 and 2020) than using the FIA dataset. Each plot within the CFI contains a radius of 24 feet and there is approximately one plot for every 40 forested acres. We used a general linear model to test the relationship between forest productivity and ECM proportion, which showed a similar pattern to the relationship between ECM proportion and productivity in the FIA dataset (Supplementary Fig. S12).

To provide additional insights into the relative importance of AM proportion and other variables in explaining variation in productivity, we fitted random forest models. We included linear and quadratic AM proportion, tree species richness, stand age, climatic variables (i.e., mean annual temperature, mean annual precipitation, and temperature seasonality), soil pH, elevation, slope, and the latitude and longitude of plots in the model. By including the latitudinal and longitudinal spatial coordinates as predictors, we addressed spatial variation in productivity by allowing the algorithm to model smooth geographical trends in productivity[75,76]. We repeated the above-mentioned random forest model by removing elevation and slope from the model and found that the overall explained variation of

productivity remained the same (Supplementary Fig. S13). Therefore, we did not include elevation and slope in the following analyses.

**Potential causal relationships between climatic conditions, soil pH, forest mycorrhizal composition, and forest productivity.** We fitted structural equation models (SEMs) to assess how climatic variables and soil pH modulated the effects of AM proportion on forest productivity in low- and high-richness plots, respectively (see Supplementary Fig. S7 for the hypothetical SEM). Again, we included linear and quadratic terms of AM proportion to model the non-linear relationships between AM tree dominance and forest productivity[77]. We included climatic variables (i.e., mean annual temperature, mean annual precipitation and temperature seasonality) and soil pH to explain AM proportion and productivity. Meanwhile, we used climatic factors to explain soil pH based on the rationale that climatic controls of litter decomposition may consequently influence soil pH[58]. We did not include tree species richness in the hypothetical SEM, because there was not necessarily a directional link between AM proportion and tree species richness. However, we did at least partly test the effects of tree species richness on the relationship between AM proportion and productivity by fitting separate SEMs for low- and high-richness plots. In both SEMs, we started with the fully specified model and performed a stepwise elimination of the path with the highest $P$-value until the Akaike Information Criterion (AIC) did not decrease.

Data manipulation and statistical analyses were done using the R platform v.4.0.3[78] and the following main packages: data.table[79], ggplot2[80], ggpubr[81], ggspatial[82], raster[83], dplyr[84], lme4[85], piecewiseSEM[86].

### Reporting summary
Further information on research design is available in the Nature Portfolio Reporting Summary linked to this article.

## Data availability
The data that support the findings of this study are available at https://doi.org/10.6084/m9.figshare.22060238. Original FIA data are available at https://apps.fs.usda.gov/fia/datamart/datamart.html. Climate data are available at the Global Climate Data·WorldClim (www.worldclim.org/). Soil pH data are available at GSDE (http://globalchange.bnu.edu.cn/research/soilw). The raw CFI dataset used for Supplementary Fig. S12 are unpublished but will be available upon reasonable request. Source data are provided with this paper.

## Code availability
R codes that support the findings of this study are available at Figshare[87] with the identifier https://doi.org/10.6084/m9.figshare.22060238. R codes for manipulating the original FIA dataset are adapted from Carteron et al.[8].

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

## Acknowledgements

S.L. acknowledges the Research Fellowship provided by the Alexander von Humboldt Foundation. S.L. and N.E. acknowledge the support of iDiv funded by the German Research Foundation (DFG- FZT 118, 202548816), and support of the Open Access Publication Fund of Leipzig University supported by the German Research Foundation within the programme Open Access Publication Costs. B.S. was supported by the University Research Priority Program "Global Change and Biodiversity" of the University of Zurich. S.F. was supported by the U.S. National Science Foundation Macrosystems Biology Program (Grant No. 2106103).

## Author contributions

S.L. and R.P.P. conceived the ideas. S.L., I.J., and B.S. analyzed the data. S.L., R.P.P., N.E., B.S., and I.J. interpreted the results. S.L. wrote the first draft of the manuscript, and all authors contributed to revisions.

## Funding

## Competing interests

The authors declare no competing interests.
