## [Peer Review File · Nature Communications]

Reviewers' Comments:

Reviewer #1:

Remarks to the Author:

This manuscript (NCOMMS-22-34695-T) reports patterns of tree richness, mycorrhizal dominance and productivity across the USA using a large public database and explores potential relationships between them using a series of statistical analyses. While the relationships between tree richness vs mycorrhizal dominance and potential environmental co-varying factors as well as biogeographical factor have already been explored in previous studies, the novelty of the paper is related to productivity. Given the current context, detailed in the last paragraph of the Discussion, this paper tackles a very important topic and represents a timely research study. Overall, I found the manuscript very well written, easy to understand and structured with elegant hypotheses. However, I have several suggestions, in particular regarding the demonstrated robustness of the study, which could potentially be improved with the additional use of variables and analyses but also regarding the contents of the methods and figures.

(1) Robustness

I believe the results could be more robust by adding key analyses and readily available covariates (see below). It doesn't diminish the value of the study and if the additional results corroborate the present results, the conclusions of the study would probably be strengthened.

The study focuses on species richness as index of alpha diversity but others measures exist and are commonly used by ecologists (e.g., Hill number framework). They allow to integrate more information than just the number of species. The Authors may want to check if their conclusions are robust using these indices. Somewhat related, the Authors use an "observed" measure of richness, however, the regional flora can greatly influence the number of tree species found in the studied sites. A null model approach would allow to account the regional species pool (e.g., Kraft et al. 2011 Science).

The Authors based the measure of forest productivity on the total aboveground live biomass divided by stand age, an estimation of mean annual increment in tree biomass. Productivity estimates are central for this study. It is not explained how stand age is evaluated. However, in the FIA database, there are also sites that were measured repeatedly and even a subset of them where remeasurement intervals are standardized (some Authors of the present study may have actually used these data previously). Therefore, would it be possible to calculate a more precise estimate of productivity from those sites and at least, see if they correlate well with the mean annual increment in tree biomass used by the Authors?

There are a number of available covariates from the FIA database that could be included in the models as they could especially affect productivity and the studied relationships. For example, elevation, slope and physiographic level. The impact of water stress is even discussed by the Authors line 198 but precipitation is not the only factor affecting it.

One of the main analyses (second hypothesis) of the manuscript relies on a threshold used to discriminate low vs high tree species richness. The authors might want to show how robust their results are to this threshold, testing for example a threshold of four, six or even three species. In any case, the Authors should probably give more details about the choice of a threshold of five species (lines 126-127). How more balanced is the number of plots by group? Also, why did the choice of two groups was retained and not three for example (low, mid and high richness)? In fact, in less than half of the ecoregion the concave-negative relationship is observed, and this somewhat weakens the second hypothesis. This number might even be lower with a threshold of four species applied. If the linear negative relationship become more frequent, it could probably change the interpretation. Although the discussion about this matter is well proposed (lines 176-189), it probably needs stronger support from the data.

The manuscript focuses on AM dominance, a quick explanation would probably help the readers to understand the reason of their choice. AM dominance is not the exact opposite of ECM dominance. Perhaps, the patterns and main models using ECM dominance should also be documented.

Because mycorrhizal type might be misidentified in some cases (Brundrett & Tedersoo, 2019 *New Phytol*), the Authors could test the robustness of their analyses to mycorrhizal misassignment. For example, analyses could be re-run by excluding the species with no information on mycorrhizal type at the species level and see if the results still hold. The Authors could use other approaches that they consider relevant to test the robustness to mycorrhizal misassignment.

Finally, regarding pH estimation, a Person's correlation of 0.60 sounds reasonable but it would be valuable if the Authors explain how the potential error in pH estimation would be conveyed in the subsequent analyses.

(2) Details of the Methods

In addition to some needed information in the Methods mentioned above, more details should be given regarding the criteria used to subset the original FIA dataset. The full dataset does not contain only natural/undisturbed forest.

Also, the codes for the manipulation of the data, the calculation of basal area and species richness, etc. are lacking.

I found the data availability statement not very clear as it is now. From this statement it is not possible to exactly know which data was used because they only provide the link to the full FIA database. The Authors should provide the dataset they used (FIA_plot_Shan Luo.csv) or the code to go from the original FIA dataset with the criteria used.

(3) Figures

Except for productivity, the maps in Figure 1 are very similar to the Figure 1 published in Carteron et al. (2022 *Nat Ecol Evo*), as such it should probably go in the supplementary as it does not really add new information. Alternatively, a bivariate map that combines productivity with mycorrhizal dominance or AM dominance could be well adapted to illustrate the main result of the manuscript.

Datapoints are highly overlapping in Figures 3, 5, S1 and S2. The Authors may want to use a point density graphical output instead.

Why is species richness not included in the theoretical SEM (Figure S3)?

Other comments:

- Statement lines 40-41 is indeed true but, mycorrhizal type is actually phylogenetically conserved and it is one of the main issues when studying the relationship between mycorrhizal type and other plant traits. Therefore, is it the best way to introduce the next paragraph about mycorrhiza?

- Perhaps, more details should be given for the second hypothesis. Any reference for the statement lines 85-86?

- References in lines 168 and 232 and corresponding rationale. I do not think phylogeny was accounted for in these studies. It is known that methods which do not account for phylogenetic relatedness or evolutionary history may overstate the relationship between traits or differences between groups (Felsenstein 1985, *Am Nat*). For example, when using spectra, is it detecting phylogenetic or taxonomic groups rather than mycorrhizal types?

- Lines 208-209, what about the importance of historical land use disturbance and other nutrient acquisition strategies such as nitrogen fixing trees (especially in eastern US)?

Reviewer #2:

Remarks to the Author:

This is a well-executed study that provides compelling evidence for a relationship between tree species richness and forest productivity that is influenced by mycorrhizal type. The work has important implications for understanding the relationship between diversity and productivity and

also for considering management options to improve forest productivity in the face of climate change. That said, the manuscript would be improved by a more careful consideration of alternative explanations for the observed relationships between proportion AM and productivity. Ultimately, what is presented here is still correlational evidence, and the authors need to be careful not to overstate their findings.

There are elements of the work that the authors should present in more detail. For example, what is the relationship between stand age and tree species richness? And how does that influence the observed AM proportion-productivity relationship? An ongoing problem with diversity-ecosystem function studies is the risk that particular species, which are more likely to be consistently present in more diverse stands, if more productive, could provide an alternative explanation for the observed relationship. Can these authors rule that possibility out? Is the increased productivity of mixed mycorrhizal forests necessarily only the result of resource complementarity? What about the role of priming of decomposition of EM litter by the presence of AM litter in mixed mycorrhizal stands? Species composition of the stands is also an important factor that was not addressed. To what extent could relationships between AM proportion and productivity observed in the different ecoregions be explained by the presence of particular suites of species and would this species composition reinforce the hypothesis advanced in this manuscript or provide an alternative explanation? The importance of longitude in their models, and the absence of a consideration of what longitude might be a proxy for, was particularly troubling. Longitude alone should not be a predictor, but it could certainly act as a proxy for the incidence of particular species or suites of species.

While overall the manuscript was well-written, the Introduction, Results and Discussion sections were unnecessarily repetitive, advancing essentially the same argument of resource partitioning and complementarity, but without expanding upon the initial presentation with detail or nuance, and also without consideration of other possible explanations. In particular, the Discussion should include a sound consideration of the assumptions and limitations of this approach, as well as some discussion of those ecoregions in which contrasting patterns were observed and why that might be the case (the manuscript did address expectations in regions of lower water availability, but these were not necessarily consistent with expected advantages of EM vs. AM species). In both the Intro and Discussion, a more concrete elaboration of how AM and EM species and stands and mixed stands might differ in their resource space, niche diversity and potential complementarity is warranted.

Responses to Reviewers' Comments

Responses to comments by Reviewer #1

Reviewer #1 (Remarks to the Author):

This manuscript (NCOMMS-22-34695-T) reports patterns of tree richness, mycorrhizal dominance and productivity across the USA using a large public database and explores potential relationships between them using a series of statistical analyses. While the relationships between tree richness vs mycorrhizal dominance and potential environmental co-varying factors as well as biogeographical factor have already been explored in previous studies, the novelty of the paper is related to productivity. Given the current context, detailed in the last paragraph of the Discussion, this paper tackles a very important topic and represents a timely research study. Overall, I found the manuscript very well written, easy to understand and structured with elegant hypotheses. However, I have several suggestions, in particular regarding the demonstrated robustness of the study, which could potentially be improved with the additional use of variables and analyses but also regarding the contents of the methods and figures.

Response: We thank the referee for this positive assessment and the helpful comments.

(1) Robustness

I believe the results could be more robust by adding key analyses and readily available covariates (see below). It doesn't diminish the value of the study and if the additional results corroborate the present results, the conclusions of the study would probably be strengthened.

Response: We thank the referee for these helpful suggestions. We have further checked the robustness of our results by: (1) adding results based on other diversity indices; (2) adding environmental covariates to our models; (3) using different cut-offs to distinguish low- from high-species richness plots; (4) adding analyses based on ECM proportion; (5) updating our dataset to include only undisturbed and natural forests, and updating mycorrhizal assignments. Our previous results hold and are now presented with additional empirical support.

The study focuses on species richness as index of alpha diversity but others measures exist and are commonly used by ecologists (e.g., Hill number framework). They allow to integrate more information than just the number of species. The Authors may want to check if their conclusions are robust using these indices. Somewhat related, the Authors use an "observed"

measure of richness, however, the regional flora can greatly influence the number of tree species found in the studied sites. A null model approach would allow to account the regional species pool (e.g., Kraft et al. 2011 Science).

Response: First, we thank the referee for pointing out that other diversity indices could be interesting to look at. Because taxonomic diversity (e.g., species richness) has been a focus of biodiversity–ecosystem functioning relationships (e.g., Liang et al. 2016 Science 354, 6309). We focus on species richness to make our results comparable to other studies. However, we have now also calculated the exponential of Shannon’s entropy index ($q = 1$) and the inverse of Simpson’s concentration index ($q = 2$), following the Hill number framework. Both indices are highly correlated with species richness and showed similar patterns as species richness. We have added the following text in the Method section (lines 383–387): “As tree species richness was highly correlated with diversity indices that account for species abundance within the Hill number framework⁶⁶, namely the exponential of Shannon’s entropy index ($q = 1$; Pearson’s correlation = 0.948, $P < 0.001$) and the inverse of Simpson’s concentration index ($q = 2$; Pearson’s correlation = 0.887, $P < 0.001$), we only presented results based on tree species richness in the main text.” And we have added that (lines 396–400) “Analyses on the effects of tree species diversity on the relationships between forest productivity and AM proportion were conducted for the exponential of Shannon’s entropy index ($q = 1$) and the inverse of Simpson’s concentration index ($q = 2$), which showed similar patterns as species richness (Fig S10&11; Table S5&6).”

Second, we agree with the referee that by accounting for regional species pools, the null model approach can improve the estimation of species diversity, especially when one wants to explain species diversity patterns. However, observed forest productivity is more likely to be influenced by the observed number of tree species in the studied sites. Moreover, by fitting ecoregions as the first term in our ANOVA, we have partly accounted for variation in productivity explained by potential variation in regional species pools. Therefore, we don’t think that the null model approach would significantly improve our prediction of productivity.

The Authors based the measure of forest productivity on the total aboveground live biomass divided by stand age, an estimation of mean annual increment in tree biomass. Productivity estimates are central for this study. It is not explained how stand age is evaluated. However,

in the FIA database, there are also sites that were measured repeatedly and even a subset of them where remeasurement intervals are standardized (some Authors of the present study may have actually used these data previously). Therefore, would it be possible to calculate a more precise estimate of productivity from those sites and at least, see if they correlate well with the mean annual increment in tree biomass used by the Authors?

Response: We have now explained how stand age is evaluated by adding the following text (lines 299-300): “*Stand age is measured using dendrochronological records⁶⁵*”. We agree with the referee that it would be even better if we could estimate productivity by repeated measures. However, we don’t think it is feasible to use repeated measures for productivity given the following reasons: 1) only some ecoregions have repeatedly-measured data (especially in the western United States), using repeated measures will significantly reduce the number of plots that can be included in the analysis; and 2) the time intervals are relatively short and not standardized (5, 7 or 10yrs). Given the enormous challenge of controlling for environmental noise in the FIA data (between different time intervals and across plots), we think that analysing repeated measures should be the focus of future studies when more repeatedly measured plots are becoming available. In contrast, estimating forest productivity as mean annual increment in tree biomass is practical, which has been used in other studies at the regional and global scales (e.g., Liang et al. 2016 Science 354, 6309; Fei et al. 2018 Nat. Commun. 9, 1-7). Moreover, it has been shown that productivity measures based on mean annual increment and period annual increment are generally consistent in a dataset compiled from global inventory forests (Liang et al. 2016 Science 354, 6309). We have added the following text to justify our productivity estimation (lines 304-310): “*We used mean annual increment in tree biomass (total aboveground live biomass divided by stand age) to estimate forest productivity^{1,26}, which enabled us to include a considerable amount of plots spreading widely across USA and to test the relatively long-term response of forest growth. Productivity measures based on mean annual increment and period annual increment have been shown to be consistent across global inventory forests¹, suggesting that mean annual increment in tree biomass could be a good proxy for productivity in our study.*”

Alternatively, we have added an extra dataset including 2,771 forest plots in the State of Indiana, USA. This dataset has repeated measures of tree basal area in five-year

intervals. We calculated periodic annual increment of basal area, which can be a proxy for productivity. We found a significant, although weak, concave-negative relationship between basal area increment and ECM proportion (Fig. S11). We think the relatively weak pattern is understandable given the limited change of basal area in short time intervals. We have added the following text (lines 387-395): *“In addition, we tested the robustness of our results with extra 2,771 managed forest plots that span ten Indiana state forests (USA) from the Continuous Forest Inventory (CFI) project of the Indiana Department of Natural Resources, Division of Forestry⁷⁴. These plots are resampled every five years, which allowed us to measure forest productivity (defined as periodic annual increment in basal area) over shorter time windows (2012 and 2020) than using the FIA data. Each plot within the CFI contains a radius of 24 feet and there is approximately one plot for every 40 forested acres. We used a general linear model to test the relationship between forest productivity and ECM proportion, which showed a similar pattern to the relationship between ECM proportion and productivity in the FIA dataset (Fig. S12).”*

Fig. S12

There are a number of available covariates from the FIA database that could be included in the models as they could especially affect productivity and the studied relationships. For example, elevation, slope and physiographic level. The impact of water stress is even discussed by the Authors line 198 but precipitation is not the only factor affecting it.

Response: In addition to variables used in the previous version of the manuscript, we have now added elevation and slope, as well as stand age, to our general linear models and random forest models. The overall patterns remained the same. Please note that with or without adding elevation and slope to the random forest models, the overall explained variation of productivity remains the same (Fig. 4 vs. Fig. S13). Therefore, we

did not include elevation and slope in our structural equation models, which focused on the causal relationships between climatic conditions, soil pH, forest mycorrhizal composition, and productivity. Accordingly, we have added the following text to the Methods section (lines 402-405): “We repeated the above random forest model by removing elevation and slope from the model, which showed that the overall explained variation of productivity remained the same (Fig. S13). Therefore, we did not include elevation and slope in the following analyses.”

One of the main analyses (second hypothesis) of the manuscript relies on a threshold used to discriminate low vs high tree species richness. The authors might want to show how robust their results are to this threshold, testing for example a threshold of four, six or even three species. In any case, the Authors should probably give more details about the choice of a threshold of five species (lines 126-127). How more balanced is the number of plots by group? Also, why did the choice of two groups was retained and not three for example (low, mid and high richness)?

Response: We are testing our second hypothesis in two ways. First, the significance of the interaction term species richness \times AM proportion [linear and quadratic terms] Table S1) shows that species richness and AM proportion had significant interactive effects on productivity. This supports our second hypothesis that the relationship between AM proportion and productivity varies with species richness. Second, we directly compare low- vs. high-richness plots by using a threshold of 5 species. As suggested by the referee, we have now further tested the robustness by using additional thresholds of four or six species. Moreover, we test the robustness by dividing plots into low- vs. intermediate- vs. high-richness plots. All these tests show the consistent pattern that AM proportion had stronger effects on productivity in less diverse plots (Fig. S6; Table S2). Therefore, we only present results based on a cut-off of 5 species and two richness groups in the main text. We have added the following text in the Methods (lines 377-382) and Results sections (lines 155-157), respectively: “Furthermore, we tested the robustness of the differences in relationships between AM proportion and productivity between low- and high-richness plots by running models with cut-offs of species richness of four and six species (Fig. S6; Table S2). We additionally test this robustness by dividing all forest plots into three groups, including low- (six species or less), intermediate- (more than six but less than 13 species), and high-richness (13 species or more) plots (Fig. S6; Table S2). Using these cut-offs of species richness generated similar patterns as a

threshold of five species.” “The stronger relationship between AM proportion and productivity in low- than high-richness plots was robust to different cut-offs of species richness (i.e., four species and six species; Fig. S6; Table S2).”

Fig. S6

In fact, in less than half of the ecoregion the concave-negative relationship is observed, and this somewhat weakens the second hypothesis. This number might even be lower with a threshold of four species applied. If the linear negative relationship become more frequent, it could probably change the interpretation. Although the discussion about this matter is well proposed (lines 176-189), it probably needs stronger support from the data.

Response: In low-richness (five or fewer species) plots, 23 out of 34 ecoregions showed concave-negative relationships, while 4 ecoregions showed linear-negative relationships (Fig. S5a). When using a cut-off of four species, 19 out of 32 ecoregions showed concave-negative relationships in low-richness (four or fewer species) plots (please see the below figure). Therefore, with a lower threshold, the number of ecoregions with concave-negative relationships slightly decreased, whereas the number of ecoregions with linear-negative relationships slightly increased. Nevertheless, the majority (~ 60%) of ecoregions did show the concave-negative relationship, which is robust to different thresholds. Therefore, we think that our second hypothesis is well supported. To interpret the linear-negative relationship, we have explicitly looked at environmental conditions, tree species diversity, and dominant tree species of ecoregions with this form of relationship. We have now thoroughly discussed potential reasons for the linear-negative relationship in the Discussion section (lines 220-240).

Low-richness (four species or less)

The manuscript focuses on AM dominance, a quick explanation would probably help the readers to understand the reason of their choice. AM dominance is not the exact opposite of ECM dominance. Perhaps, the patterns and main models using ECM dominance should also be documented.

Response: We understand the point of the referee that AM dominance is not the exact opposite of ECM dominance due to the presence of other mycorrhizal strategies, although these are rare in our dataset. We have added the following text (lines 87-91)

“To quantify forest mycorrhizal composition, we calculated AM (or ECM) proportion as the total basal area of AM (or ECM) tree species divided by the total stand basal area. The patterns of ECM and AM proportions are essentially mirror images (Fig. S1c&d); therefore, we used AM proportion throughout the text, with increased AM proportion indicating increased dominance of AM tree species.” We are now showing the relationship between productivity and ECM tree dominance in the supplement (Fig. S4; Appendix S1 for statistical results).

Fig. S4

Because mycorrhizal type might be misidentified in some cases (Brundrett & Tedersoo, 2019 New Phytol), the Authors could test the robustness of their analyses to mycorrhizal misassignment. For example, analyses could be re-run by excluding the species with no information on mycorrhizal type at the species level and see if the results still hold. The Authors could use other approaches that they consider relevant to test the robustness to mycorrhizal misassignment.

Response: In the previous version of the manuscript, we used mycorrhizal assignment from Jo et al. (2019 Sci. Adv. 5, eaav6358). To be more precise in mycorrhizal assignment, we have now updated our mycorrhizal assignment based on a recently published database (Steidinger et al. 2019 Nature 569, 404-408), which provides species-level mycorrhizal strategies. About 83.3% of the species in our study are included in this database. For the remaining species, we extracted mycorrhizal information from a second database (Soudzilovskaia et al. 2020 New Phytol. 227, 955-966), which assigns mycorrhizal strategies at the genus level. Therefore, we think that the possibility of

mycorrhizal misassignment due to mismatch between genus- and species-level assignment would be reasonably low in our study, at least based on currently available literatures. We have edited the following text in the Methods section (lines 323-326):
“The mycorrhizal strategy for each tree species was assigned based on a recently published database⁵², which provides species-level mycorrhizal assignment; 314 out of the 377 species in our study were found in this database. For the remaining 63 species, we extracted genus-level mycorrhizal assignment from another database⁶⁸.”

Finally, regarding pH estimation, a Person’s correlation of 0.60 sounds reasonable but it would be valuable if the Authors explain how the potential error in pH estimation would be conveyed in the subsequent analyses.

Response: We thank the referee for pointing this out. We have added the following text on lines 348-351: *“We note that the potential error in pH estimation is likely to underestimate or overestimate the effects of soil pH on productivity. However, this uncertainty applies in the same way to analyses for low- and high-richness plots and thus is unlikely to affect any of our findings or conclusions.”*

(2) Details of the Methods

In addition to some needed information in the Methods mentioned above, more details should be given regarding the criteria used to subset the original FIA dataset. The full dataset does not contain only natural/undisturbed forest.

Response: We have now updated our final dataset by excluding unnatural/disturbed forest plots, following the criteria of Carteron et al. (2022 Nat. Eco. Evo. 6, 370-374). Our final dataset now includes 74,563 plots, and we find the same patterns as in the previous version of the manuscript. We have clarified this by adding the following text on lines 300-302: *“We subset the original FIA dataset following the protocol of Carteron et al.⁸, which only kept census data from the most recent year for a given plot and from natural and undisturbed forests following standardized census methods.”*

Also, the codes for the manipulation of the data, the calculation of basal area and species richness, etc. are lacking.

Response: R codes for manipulating the original FIA dataset are adapted from Carteron et al. (2022 Nat. Eco. Evo. 6, 370-374). We have added code for the calculation of basal area and species richness.

I found the data availability statement not very clear as it is now. From this statement it is not possible to exactly know which data was used because they only provide the link to the full FIA database. The Authors should provide the dataset they used (FIA_plot_Shan Luo.csv) or the code to go from the original FIA dataset with the criteria used.

Response: We have edited the Data availability statement as: “*The data that support the findings of this study are available at (provide DOI upon publication).*”

(3) Figures

Except for productivity, the maps in Figure 1 are very similar to the Figure 1 published in Carteron et al. (2022 Nat Ecol Evo), as such it should probably go in the supplementary as it does not really add new information. Alternatively, a bivariate map that combines productivity with mycorrhizal dominance or AM dominance could be well adapted to illustrate the main result of the manuscript.

Response: We thank the referee for the suggestion. We have moved this figure to the supplement as Fig. S1.

Datapoints are highly overlapping in Figures 3, 5, S1 and S2. The Authors may want to use a point density graphical output instead.

Response: We thank the referee for this suggestion. We have changed the shape and transparency of data points to improve our figures.

Why is species richness not included in the theoretical SEM (Figure S3)?

Response: We did not include species richness in the theoretical SEM for two main reasons. First, although mycorrhizal fungi can influence plant community diversity, there is not necessarily a directional link between AM proportion and tree species richness. Second, consistent with our second hypothesis, low- and high-richness plots showed different relationships between AM proportion and productivity (i.e., the interaction term of richness and AM proportion was statistically significant in general linear models). To complement our general linear models, we further used SEMs to explore how soil pH and climatic variables modulated the effects of AM proportion on productivity in low- vs. high-richness groups. That said, we aimed to test the overall effects of AM proportion on productivity, and we have at least partly tested the effects of richness on the AM proportion-productivity relationship by constructing separate

SEMs for low- vs. high- richness plots. We have added the following text to the figure legend to clarify that this hypothetical SEM applies to both low- and high-richness plots (lines 828-829): *“The SEMs were constructed separately for low- and high-richness plots”*. In addition, we have added the following text in the Methods section (lines 416-420): *“We did not include tree species richness in the hypothetical SEM, because there was not necessarily a directional link between AM proportion and tree species richness. However, we did at least partly test the effects of tree species richness on the relationship between AM proportion and productivity by fitting separate SEMs for low- and high-richness plots.”*

Other comments:

- Statement lines 40-41 is indeed true but, mycorrhizal type is actually phylogenetically conserved and it is one of the main issues when studying the relationship between mycorrhizal type and other plant traits. Therefore, is it the best way to introduce the next paragraph about mycorrhiza?

Response: We have now deleted these lines and addressed the phylogeny-related issue in the Discussion section by adding the following text (lines 261-263): *“Third, whether our findings emerge from differences in mycorrhizal traits (between AM and ECM fungi) vs. differences in suites of plant traits (between AM- and ECM associating tree species) requires further study.”*

- Perhaps, more details should be given for the second hypothesis. Any reference for the statement lines 85-86?

Response: We have added references to our second hypothesis. It now reads as follows (lines 96-99): *“Second, we hypothesized that the effects of mycorrhizal mixtures on forest productivity would be stronger in species-poor than in species-rich stands, because the latter may make up for a lack of resource partitioning via mycorrhizal associations with other functional-diversity strategies³¹. Alternatively, ECM and AM tree species as groups may overlap more strongly in resource use if there are more species in each group³²⁻³⁴, again causing weaker effects of mycorrhizal associations in more than less species-rich stands.”*

- References in lines 168 and 232 and corresponding rationale. I do not think phylogeny was

accounted for in these studies. It is known that methods which do not account for phylogenetic relatedness or evolutionary history may overstate the relationship between traits or differences between groups (Felsenstein 1985, Am Nat). For example, when using spectra, is it detecting phylogenetic or taxonomic groups rather than mycorrhizal types?

Response: We agree with the referee that the cited studies, as well as some other studies on this topic, do not account for phylogeny when comparing AM and ECM tree species; therefore, they may have overstated the differences between the two groups.

Accordingly, we have toned down our wording when inferring that the potential differences between AM and ECM tree species may be the driver of the observed pattern. We have addressed this issue by adding the following text to the Discussion section (lines 257-260): “*Second, we acknowledge that the ecophysiological differences between mycorrhizal associations might have been overestimated in previous studies, which often, though not always²⁶, did not correct for phylogenic correlation among tree species.*” In terms of the spectra, we have revised the text on lines 283-285: “*In addition, emerging methods based on remote sensing show promise for generating global estimates of distributions of mycorrhizal associations⁶³, ...*”

- Lines 208-209, what about the importance of historical land use disturbance and other nutrient acquisition strategies such as nitrogen fixing trees (especially in eastern US)?

Response: We have deleted this sentence.

Responses to comments by Reviewer #2

Reviewer #2 (Remarks to the Author):

This is a well-executed study that provides compelling evidence for a relationship between tree species richness and forest productivity that is influenced by mycorrhizal type. The work has important implications for understanding the relationship between diversity and productivity and also for considering management options to improve forest productivity in the face of climate change. That said, the manuscript would be improved by a more careful consideration of alternative explanations for the observed relationships between proportion AM and productivity. Ultimately, what is presented here is still correlational evidence, and the authors need to be careful not to overstate their findings.

Response: We thank the referee for pointing out that our study is well-executed and important. We agree with the referee that, as with other observational studies, our study can only provide correlational evidence. Therefore, we have been careful with our wording to avoid overstating our findings. And we have added alternative explanations for the observed relationship between AM proportion–productivity. Below, we provide a point-to-point response to major comments, as well as comments added in the previous version of the text (“2_reviewer_attachment_1_1666590412_convrt”).

There are elements of the work that the authors should present in more detail. For example, what is the relationship between stand age and tree species richness? And how does that influence the observed AM proportion-productivity relationship?

Response: We thank the referee for pointing this out. We now add stand age to our ANOVAs (Table S1, S2, S5, S6) and random forest models. Importantly, our previous results hold. We are also showing the relationship between stand age and AM proportion in the supplement (Fig. S9). There is only a weak, negative relationship between stand age and AM proportion. In addition, we have tested the relationship between AM proportion and productivity for young and old stands separately and found similar patterns. Therefore, we think that the observed concave-negative relationship between AM proportion and productivity was not likely confounded by the relationship between AM proportion and stand age. We have added the following text in the Methods section (lines 334-336): “*There is only a weak, negative relationship (Fig. S9; $P < 0.001$, $R^2 = 0.002$) between AM proportion and stand age.*”

Fig. S9

The mean stand age is 78 years,

which was used as a threshold for grouping young vs. old stands.

An ongoing problem with diversity-ecosystem function studies is the risk that particular species, which are more likely to be consistently present in more diverse stands, if more productive, could provide an alternative explanation for the observed relationship. Can these authors rule that possibility out? Is the increased productivity of mixed mycorrhizal forests necessarily only the result of resource complementarity? What about the role of priming of decomposition of EM litter by the presence of AM litter in mixed mycorrhizal stands? Species composition of the stands is also an important factor that was not addressed. To what extent could relationships between AM proportion and productivity observed in the different ecoregions explained by the presence of particular suites of species and would this species composition reinforce the hypothesis advanced in this manuscript or provide an alternative explanation?

Response: The referee has three main concerns. In our revised manuscript, we have addressed them as detailed below.

First, the referee points out that it is a common challenge of biodiversity-ecosystem function studies to disentangle sampling or selection effects from species complementarity effects. The referee is right that we cannot rule out the possibility that productive species can disproportionately contribute to productivity. We have added the following text to the Discussion section (lines 253-257): “First, our study cannot exclude the possibility that particular productive species can contribute to the observed relationship between AM tree dominance and productivity (the so-called ‘selection effect’⁶⁵). However, the consistent concave-negative relationship between AM tree dominance and productivity

in a majority of ecoregions demonstrate that the effects of forest mycorrhizal composition on productivity may be quite general.”

Second, the referee thinks that the increased productivity of mixed mycorrhizal forests may result from priming of decomposition of EM litter by the presence of AM litter. If this is the case, we would expect that ECM litters can decompose faster in AM or mixed stands than in home stands. However, there is little evidence for this in the few studies that have investigated this topic, as ECM litters tend to decay slowly regardless of the soil environment (Jacobs et al. 2018 J. Ecol. 106, 502-513; Midgley et al. 2015 J. Ecol. 103, 1454-1463). Likewise, it has been shown that the decomposition rate of ECM litter was not influenced by the relative dominance of AM or ECM trees or soil inorganic N concentration (Midgley et al. 2015 J. Ecol. 103, 1454-1463). We have added the following text to the Discussion section to discuss alternative explanations (lines 193-203): *“In addition to resource partitioning, mixed-mycorrhizal stands may be productive owing to resource enrichment and biotic feedbacks⁴¹. While ECM litters have been shown to decay more slowly regardless of the soil environment^{42,43}, herbaceous AM plants in the understory of an ECM-dominated stand can accelerate soil organic matter decomposition⁴⁴. If AM trees facilitate decomposition in a similar way, mixed-mycorrhizal stands may have more nutrients in circulation compared to stands dominated by a single mycorrhizal type. Moreover, it is plausible that mixing mycorrhizal strategies could have non-nutritional benefits, given that AM and ECM tree species generally differ in biotic interactions with other trophic levels⁴⁵. For instance, the mycorrhizal fungal partners may complement each other in protection against pathogens and herbivores¹⁷. These differences between AM and ECM tree species may lead to complementarity and enhance productivity in stands with mixed mycorrhizal strategies⁴¹.”*

Third, the referee queries about whether different relationships between AM proportion and productivity can be attributed to different communities across different ecoregions. We think that by fitting ecoregion in the beginning of our hierarchical ANOVA, we have partly accounted for spatial heterogeneity across ecoregions, including potential differences in species composition. After excluding the effects of ecoregion, AM proportion and the interaction between AM proportion and ecoregion still showed significant effects on productivity. This suggests that the different AM proportion–productivity relationships across ecoregions are not solely due to the

presence of different suites of species. Moreover, the concave-negative relationship was found in the majority (i.e., 23 out of 34) of ecoregions, covering a wide range of areas with different species compositions (Knott et al. *Global Ecol. Biogeogr.* 2020 00, 1-16). This can, to some extent, suggest the generality of the concave-negative relationship between productivity and AM tree dominance. In contrast, only four ecoregions showed the linear-negative relationship. Those ecoregions have relatively low mean annual precipitation, high elevation, and low species richness. They are dominated by a few drought-tolerant ECM tree species (i.e., *Pinus*) and shrubby AM tree species (i.e., *Prosopis*, *Juniperus*). In this case, the higher productivity in ECM- than in AM-dominated stands is likely due to the presence of particularly drought-tolerant species. We have thoroughly discussed this now on lines 220-240. For example, we have added the following text (lines 237-240): “*Thus, we suspect that the higher productivity in ECM- than in AM-dominated stands could be attributed to the effects of particular suites of species (i.e., productive ECM tree species or unproductive AM tree species), which could play stronger roles in ecoregions with fewer tree species.*”

The importance of longitude in their models, and the absence of a consideration of what longitude might be a proxy for, was particularly troubling. Longitude alone should not be a predictor, but it could certainly act as a proxy for the incidence of particular species or suites of species.

Response: As stated on lines 135-136, “*Including latitude and longitude allowed the algorithm to account for spatial variation in productivity*” in the random forest models. Rather than suggesting that longitude *per se* has ecological effects on productivity, the statistical significance of longitude in the model indicates that there is a geographical pattern in productivity (and this has been shown in another study using the FIA dataset: Craven et al. 2020 *Glob. Ecol. Biogeogr.* 29, 1940-1955). Therefore, we would like to clarify that we did not aim to test how longitude *per se* influences productivity, but to test the relative importance of AM proportion and other variables in explaining variation in productivity after accounting for spatial variation in productivity. While we agree with the referee that longitude could act as a proxy for the incidence of particular species or suites of species, it could also be a proxy for other unknown variables, such as environmental variables, that vary spatially and can lead to spatial variation in productivity. Regardless of what longitude could be a proxy for, our results show that after accounting for spatial variation in productivity by including latitude and longitude

as explanatory variables, AM proportion still explained a significant amount of variation in productivity. Nevertheless, we are discussing now how the potential variation in species composition across ecoregions may influence the relationship between AM proportion and productivity in the Discussion section. For example, we have added the following text (lines 237-240): *“Thus, we suspect that the higher productivity in ECM- than in AM-dominated stands could be attributed to the effects of particular suites of species (i.e., productive ECM tree species or unproductive AM tree species), which could play stronger roles in ecoregions with fewer tree species.”*

While overall the manuscript was well-written, the Introduction, Results and Discussion sections were unnecessarily repetitive, advancing essentially the same argument of resource partitioning and complementarity, but without expanding upon the initial presentation with detail or nuance, and also without consideration of other possible explanations. In particular, the Discussion should include a sound consideration of the assumptions and limitations of this approach, as well as some discussion of those ecoregions in which contrasting patterns were observed and why that might be the case (the manuscript did address expectations in regions of lower water availability, but these were not necessarily consistent with expected advantages of EM vs. AM species). In both the Intro and Discussion, a more concrete elaboration of how AM and EM species and stands and mixed stands might differ in their resource space, niche diversity and potential complementarity is warranted.

Response: We thank the referee for the suggestion. We have now improved our Introduction section by elaborating in detail about how different mycorrhizal associations may complement each other in the acquisition of limiting nutrients (lines 52-64). We have also improved our Discussion with regards to four main aspects. First, we have explained how mixing AM and ECM tree species may increase the overall resource exploitation in stands with mixed mycorrhizal strategies (lines 182-192). Second, we have added alternative explanation for the positive effects of mixing AM and ECM tree species on productivity (lines 193-203). For example, we have added the following text: *“In addition to resource partitioning, mixed-mycorrhizal stands may be productive owing to resource enrichment and biotic feedbacks⁴¹.”* Third, we have thoroughly discussed potential reasons for the linear-negative relationship between AM proportion and productivity on lines 220-240. We did so by referring to potential effects of dominant tree species, species diversity, and environmental conditions in those ecoregions. Lastly, we have acknowledged the assumption and limitation of our

approach (lines 252 and lines 267). For instance, we have further clarified our assumption on lines 286-289: *“Lastly, while our analysis has lumped diverse plants and fungi into AM versus ECM functional groups, we acknowledge that different tree species and mycorrhizal fungi within and across functional groups have diverse ecological strategies⁶² and that exploring this diversity may further improve predictions of forest productivity⁵¹.”*

L71-72. So if stand age is confounded with tree diversity or AM:EM, then relationship to productivity could be solely correlative. Given this measure of productivity, I think it's important to indicate how stand age varies with ecoregion and AM proportion. Differences could be driven by the denominator as much as the numerator.

Response: As responded above, we now added stand age in our ANOVAs (Table S1, S2, S5, S6) and random forest models. And we are now showing the relationship between stand age and AM proportion in the supplement (Fig. S9). There is only a weak, negative relationship between stand age and AM proportion ($R^2=0.002$). Therefore, it is not likely that the relationship between productivity and AM proportion could be confounded by the relationship between stand age and AM proportion. However, the relationship between stand age and tree diversity would be less relevant for testing the relationship between productivity and AM proportion. In fact, stand ages can even vary within ecoregions (e.g., ranging from 14 to 360 years in Pacific Lowland Mixed Forest). Therefore, it is not likely that the different patterns (i.e., concave-negative relationships in 26 ecoregions vs. linear-negative relationships in 4 ecoregions vs. non-significant relationships in 4 ecoregions) observed in different ecoregions were mainly due to their differences in stand ages.

L81: The shape of this curve would not be dictated by this hypothesis.

Response: In the Introduction section, we suggest that our hypothesis would predict a concave-negative relationship between AM tree dominance and productivity. However, we understand the point of the referee that other possible processes may also lead to such a pattern. Therefore, we have added alternative explanations for the concave-negative relationship in the Discussion section (lines 193-203): *“In addition to resource partitioning, mixed-mycorrhizal stands may be productive owing to resource enrichment and biotic feedbacks⁴¹. While ECM litters have been shown to decay more slowly regardless of the soil environment^{42,43}, herbaceous AM plants in the understory of an*

ECM-dominated stand can accelerate soil organic matter decomposition⁴⁴. If AM trees facilitate decomposition in a similar way, mixed-mycorrhizal stands may have more nutrients in circulation compared to stands dominated by a single mycorrhizal type. Moreover, it is plausible that mixing mycorrhizal strategies could have non-nutritional benefits, given that AM and ECM tree species generally differ in biotic interactions with other trophic levels⁴⁵. For instance, the mycorrhizal fungal partners may complement each other in protection against pathogens and herbivores¹⁷. These differences between AM and ECM tree species may lead to complementarity and enhance productivity in stands with mixed mycorrhizal strategies⁴¹.”

L84: alternatively, with increasing diversity may come an increasing likelihood that different species compete for resources in the same way.

Response: We are not sure what the referee means by ‘alternatively, with increasing diversity may come an increasing likelihood that different species compete for resources in the same way’ as alternative to ‘*Second, we hypothesized that the effects of mycorrhizal mixtures on forest productivity would be stronger in species-poor than in species-rich stands, because the latter may make up for a lack of resource partitioning via mycorrhizal associations with other functional-diversity strategies*’. While we agree that with increasing diversity different species should compete more strongly for resources in the same way, but we don’t see how this would explain the weaker effects of mycorrhizal mixtures. Nevertheless, we think it is likely that there would be more overlap in resource use between ECM and AM tree species as two groups, because with more species in each category there is a greater likelihood that some ECM tree species are similar to AM tree species and the other way around. We have revised our second hypothesis accordingly (lines 97-102): “*Second, we hypothesized that the effects of mycorrhizal mixtures on forest productivity would be stronger in species-poor than in species-rich stands, because the latter may make up for a lack of resource partitioning via mycorrhizal associations with other functional-diversity strategies³¹. Alternatively, ECM and AM tree species as groups may overlap more strongly in resource use if there are more species in each group³²⁻³⁴, again causing weaker effects of mycorrhizal associations in more than less species-rich stands.*”

L92: for the first three main effects? Why these?

Response: We justify this on lines 110-113: “..., *interactions between AM proportion and ecoregion, and interactions between AM proportion and richness as explanatory variables. The interaction terms were used to test whether relationships between AM proportion and productivity change across ecoregions or the species-richness gradient.*”

L97: but these are captured in the ecoregions.

Response: We agree with the referee that including ecoregion in the model has already captured, at least some, variation in environmental conditions. However, there should be additional environmental variations across plots within ecoregions. As our study is testing the relationship between AM proportion and productivity using a plot-level dataset, we think it is better to add plot-level environmental covariates to further control for environmental variations. This approach has been widely used (e.g., Cation et al. 2022 *Nat. Eco. Evo.* 6, 370-374; Averill et al. 2022 *Nat. Eco. Evo.* 6, 375-382), as well as supported by the first referee by suggesting us to include a few more environmental variables.

L103: Clarify whether most ecoregions are located in the eastern UAS, most ecoregions with this relationship between AM proportion and productivity are in the eastern USA, or both.

Response: We have revised this sentence on lines 124-126: “*In particular, 26 out of 34 (~76.5%) ecoregion-level analyses yielded significantly concave-negative relationships between forest productivity and AM proportion (Fig. 2).*”

L105: To draw this conclusion, it would be important to establish that this relationship isn't being driven instead by particular tree species that are boosting the productivity, in which case the relationship may or may not be due to mycorrhizal type.

Response: We have toned down our wording and revised this sentence (lines 128-130): “*Overall, these results showed that forests with a mixture of both mycorrhizal strategies tended to have higher productivity than forests dominated by either ECM or AM tree species*”. Accordingly, we have added the following text in the Discussion section (lines 215-219): “*First, our study cannot exclude the possibility that particular productive species can contribute to the observed relationship between AM tree dominance and productivity (the so-called ‘selection effect’⁶⁵). However, the consistent concave-negative relationship between AM tree dominance and productivity in a majority of ecoregions*”

demonstrate that the effect of forest mycorrhizal composition on productivity may be quite general.”

L112: longitude per se should have no influence on productivity.

Response: As responded above, we included latitude and longitude of plots in the random forest model to account for potential geographical trends in productivity (lines 135-136). Therefore, we did not suggest that longitude per se has any ecological effects on productivity.

L113-114: East-west is unlikely to be a spatial component but rather species composition.

Response: We have deleted this sentence to avoid misleading the readers, because we do not aim to emphasize the geographical patterns in productivity. Instead, we would like to show that even after accounting for spatial variation in productivity by including longitude as an explanatory variable, AM proportion still explained a significant amount of variation in productivity. In addition to species composition, we think that longitude could be a proxy for other unknown variables, such as environmental variables that are not included in our study. The revised text is on lines 135-138: *“Including latitude and longitude allowed the algorithm to account for spatial variation in productivity. After accounting for spatial variation in productivity, species richness was the most important predictor of productivity, with linear and quadratic AM proportion showing comparable importance (Fig. 3).”*

L174: But the role of particularly productive tree species that dominate certain types of forests can't be ruled out.

Response: We thank the referee for pointing this out. We have added the following text (lines 253-255): *“First, our study cannot exclude the possibility that particular productive species can contribute to the observed relationship between AM tree dominance and productivity (the so-called ‘selection effect’⁶⁵).”*

L179-181: I struggle to see the validity of this argument. AM-dominated forests lack the accumulated organic matter that provides ready organic sources for nitrogen and phosphorus. The resource space differs, which should generate more competition among species for mineral nutrients, while in mixed forests, the resource space is effectively greater.

Response: We understand the point of the referee that there should be weaker competition among species in mixed forests than AM-dominated forests, which is consistent with our first hypothesis and supported by the results. However, we additionally hypothesize that this difference between forests dominated by single vs. mixed mycorrhizal strategies would be more pronounced in species-poor than species-rich stands. Because we expect that in diverse stands with mixed AM and ECM tree species, species competition would be as strong as that in diverse stands dominated by AM or ECM trees. We have revised the text accordingly (lines 208-215): *“Particularly, with more species within the groups of ECM and AM tree species, there is greater likelihood that some ECM and AM tree species may overlap in resource use. This may result in weaker effects of mycorrhizal associations in more than less species-rich stands. However, other types of functional diversity among species may counter this and enhanced complementary and facilitative resource extraction in diverse communities⁴⁹. It is thus likely that a large number of tree species can exploit the total resource space via several mechanisms⁵⁰ and in this case, the addition of mixed mycorrhizal strategies may have little benefit for increasing resource space exploited (Fig. 1c).”*

181-184: A simpler argument is that more diverse stands are older, so productivity as you have measured is necessarily lower (since you are dividing biomass by stand age).

Response: This comment by the referee would imply that productivity would be lower in more diverse stands than in less diverse stands, which, however, wasn't supported by our results. In contrast, productivity increased with increasing tree species richness (Fig. S3).

Fig. S3

L196-198: I think this deserves a bit more consideration. If ECM tree species are more drought tolerant, then what would be the expectation for the AM proportion-productivity relationship in low MAP sites and are your findings consistent with that?

Response: Indeed, we found linear-negative relationships between AM proportion and productivity in four ecoregions, where MAP was relatively low. There is literature suggesting that ECM tree species are in general more drought-tolerant than AM tree species (e.g., Brzostek et al. 2014 Glob. Chang. Biol. 20, 2531-2539), which is consistent with the observation that ECM tree species can dominate dry climates (Steidinger et al. 2019 Nature 569, 404-408). Pines are dominant species in these dry ecoregions, suggesting that drought-tolerant ECM tree species can lead to the high productivity in ECM-dominated stands. Nevertheless, some AM tree species (mainly *Prosopis* and *Juniperus*) are also thought to be drought tolerant (but see Johnson et al. 2018 Plant, Cell & Environment, 41, 576-588) and can become dominant in dry ecoregions. But those AM species are shrubs and/or small-size trees with presumably low aboveground productivity, which may have led to the low productivity in AM-dominated stands. We have thoroughly discussed this on lines 220-240.

L204-205: This should be spelled out more clearly for those readers less familiar with the relationships between mycorrhizal type, leaf litter quality and nutrient cycling.

Response: We have added the following text (lines 245-246): “*Note that trees with different mycorrhizal associations can also differentially modify soil pH via distinct litter quality and nutrient cycling processes*^{68,69}.”

L206-207: maybe there is a typo here? How does a shift ripple up? Clarity.

Response: We have revised the text on lines 248-251: “*Nevertheless, our study suggests that global environmental changes may shift forest mycorrhizal composition and consequently influence forest productivity, which can have greater impacts on species-poor than species-rich forests.*”

L209-210: Would Lilleskov’s work and others looking at gradients of N deposition in ECM forests shed light on this?

Response: We thank the referee for the suggestion. We have deleted this sentence.

L216: two is not exactly ‘a diversity’. And the data here are correlative. More work needs to be done to establish the driver of the productivity.

Response: We thank the referee for pointing this out. We have revised the text on lines 271-272: “*In conclusion, we show that forests with mixed mycorrhizal strategies have higher productivity than that dominated by single mycorrhizal strategy across the contiguous USA...*”

L:217-220: the wording here isn’t quite right. Your argument is that highly diverse forests will have high productivity regardless of proportion AM, because there will be high complementarity.

Response: We thank the referee for the suggestion. We have revised this sentence on lines 273-274: “*Forests with high tree diversity achieve high productivity regardless of tree mycorrhizal strategies, probably via...*”.

L500-502: ‘In our illustration, the boxes represent the total available resource space, and circles represent the resource space occupied by tree species (orange circles represent AM tree species, green circles represent ECM tree species).’ Comments: ‘These boxes don’t adequately represent the resource space occupied according to your hypotheses, because if AM and ECM species occupied different niches, then there should be empty niches in stands with only one mycorrhizal type’.

Response: As stated in the legend (lines 654-655), “*the boxes represent the total available resource space, circles represent the resource space occupied by tree species*”; therefore, the space outside the circles would represent empty niches. We have now made the ‘empty niches’ more recognizable by changing the colour from white to grey and have added the followed text on lines 656-657: “*... and grey areas represent unconsumed resources*”.

L581: Fig. S3. Here we can see most clearly the potential for this relationship between AM proportion and productivity to be driven instead by floristic differences.

Response: This comment refers to Fig. S5 in the revised manuscript. This point of the referee is not very clear to us. But we think the referee means that different (concave-negative vs. linear-negative) relationships between AM proportion and productivity are potentially driven by different species compositions across different ecoregions. We generally agree with the referee on this. In our study, 26 out of 34 ecoregions showed

concave-negative relationships, while four ecoregions showed linear-negative relationships. We think that the consistent concave-negative relationship between AM tree dominance and productivity in a majority of ecoregions indicates the generality of the effects of forest mycorrhizal composition on productivity, regardless of potentially different species pools across ecoregions. In contrast, it is likely that the other four ecoregions contain specific sets of species that can predominantly drive the linear-negative relationship. Specifically, we suspect that the declining pattern of productivity with increasing AM tree dominance was driven by particularly drought-tolerant tree species, either ECM tree species that are doing well despite harsh environments (e.g., dry) or shrubby AM species with presumably low aboveground productivity. We have added the following text in the Discussion section (lines 230-240): *“Additionally, the negative correlation between AM dominance and productivity may relate to the smaller regional species pool in these arid regions (Table S4). In one eco-region (Black Hills coniferous forest), the ECM-associating Pinus comprised 97% of the total basal area (Table S3), such that this relatively drought-tolerant species was mostly responsible for the greater productivity in ECM- than AM-dominated stands. Likewise, the dominant AM tree species (i.e., Prosopis, Juniperus; Table S3), though often presumed to be drought tolerant^{59,60}, often occur as shrubs or small-sized trees with presumably low aboveground productivity. Thus, we suspect that the higher productivity in ECM- than in AM-dominated stands could be attributed to the effects of particular suites of species (i.e., productive ECM tree species or unproductive AM tree species), which could play stronger roles in ecoregions with fewer tree species.”*

Reviewers' Comments:

Reviewer #1:

Remarks to the Author:

Dear Editor,

I am satisfied with the revisions made to the manuscript, which have addressed most of my concerns. The new analyses have provided further evidence to support the study, particularly through the use of the CFI. Please make sure the data and scripts used for this additional analysis are shared.

Furthermore, the Authors actually recognized the importance of the species pool (for example lines 231-232) but argue against the use of a null model, which seems contradictory. However, the following discussion (line 232 and onwards) is very interesting.

Similarly, I would have liked to see an explanation as to why the physiographic levels were not taken into account.

Additionally, I have noticed an inconsistency between the reference numbers cited in the revised manuscript and in the rebuttal. For example, lines 323-326 refs 58 and 73 vs 52 and 68 in the rebuttal).

I think that for completeness, the authors should explain that the genus-level information is based on a 2/3 ratio from Soudzilovskaia et al. (2020).

Lastly, I am not sure of the journal's policies, however, datapoints are still highly overlapping in several plots. Also, I suggest that the authors consider including confidence intervals in the plots when possible.

Reviewer #2:

Remarks to the Author:

In my view, the authors did a very thorough job of addressing the concerns the reviewers had with the initially submitted manuscript. The key finding of greater forest productivity for sites with mixed AM-ECM tree species was well-supported and potentially confounding factors were appropriately considered. The revisions greatly strengthened the introduction, the interpretation of the results and the conclusions, as well as the overall flow of the manuscript. The resulting revised manuscript draws well-reasoned conclusions from a large dataset, with interesting implications for forest ecologists and managers.

REVIEWERS' COMMENTS

Reviewer #1 (Remarks to the Author):

Dear Editor,

I am satisfied with the revisions made to the manuscript, which have addressed most of my concerns. The new analyses have provided further evidence to support the study, particularly through the use of the CFI. Please make sure the data and scripts used for this additional analysis are shared.

Response: We thank the referee for this positive assessment. Regarding the CFI dataset, we have provided source data and R scripts used to generate the corresponding Supplementary Fig. S12, which should be useful for the readers to replicate all data points and the curve in Supplementary Fig. S12. We prefer not to publish the raw data, because another manuscript is in preparation.

Furthermore, the Authors actually recognized the importance of the species pool (for example lines 231-232) but argue against the use of a null model, which seems contradictory. However, the following discussion (line 232 and onwards) is very interesting.

Response: We understand that the referee suggested using a null model to estimate tree species richness to account for the potential *indirect* effects of regional species pool on productivity via mediating tree species richness. We prefer not to use of this approach because of two main reasons. First, we think that “observed forest productivity is more likely to be influenced by the observed number of tree species in the studied sites.” We doubt that using estimated species richness would significantly improve prediction of productivity, compared to observed species richness. Second, we have shown the robustness of the effects of tree species diversity on productivity by providing results based on other tree diversity indices (i.e. Hill numbers).

The referee is right that we did suspect that regional species pools may be important for the negative relationship between AM tree dominance and productivity in four (~11%) ecoregions, which, however, is not contradictory to our above argument. On the one hand, we suspect that particular suits of species may have led to higher productivity in ECM- than in AM-dominated stands in these ecoregions (lines 231-240 in the previous version of the manuscript or lines 237-247 in the current version of the manuscript). On the other hand, we found consistent concave-negative relationships between AM dominance and productivity in ~77% of the ecoregions, suggesting that this pattern may be quite general and independent of species pool. Nevertheless, we had addressed the potential *direct* effects of species pool on productivity by fitting ecoregions as the first term in our ANOVAs, assuming that different ecoregions have different species pools, which can partly account for variation in productivity explained by potential variation in regional species pools. This is now more clearly explained in the revised manuscript with the following text (lines 377-380): “*We addressed the potential effects of regional species pool on productivity by fitting ecoregions as the first term in the GLMs, assuming that different ecoregions have different species pools, which can partly account for variation in productivity explained by potential variation in regional species pools.*”

Similarly, I would have liked to see an explanation as to why the physiographic levels were not taken into account.

Response: We followed the reviewer's suggestion and added the physiographic level of each plot to our final dataset. We have tested the effects of physiographic level on productivity by adding it as an additional covariate in our ANOVAs and presented the results in Supplementary Table S5. Since it only explains a small fraction of variation in productivity, we exclude it from final models. We now add the following text in the Methods section (lines 380-383): *“Adding physiographic class in the GLMs hardly reduced the sum of squares of residuals, because it only explained a small fraction of variation in productivity (Supplementary Table S5). Therefore, we excluded physiographic class from all models; this did not influence our main findings and conclusions.”*

Additionally, I have noticed an inconsistency between the reference numbers cited in the revised manuscript and in the rebuttal. For example, lines 323-326 refs 58 and 73 vs 52 and 68 in the rebuttal).

Response: We thank the referee for pointing this out. The reference numbers 52 and 68 in the following sentences of the previous rebuttal should be replaced by 58 and 73, respectively.

“The mycorrhizal strategy for each tree species was assigned based on a recently published database⁵², which provides species-level mycorrhizal assignment; 314 out of the 377 species in our study were found in this database. For the remaining 63 species, we extracted genus-level mycorrhizal assignment from another database⁶⁸.”

I think that for completeness, the authors should explain that the genus-level information is based on a 2/3 ratio from Soudzilovskaia et al. (2020).

Response: We thank the referee for the suggestion. We have edited the following text in the Methods section (lines 333-334): *“For the remaining 63 species, we extracted genus-level mycorrhizal assignment from another database⁷³, which assigned genus-level information when > 67% of the observations were consistent.”*

Lastly, I am not sure of the journal's policies, however, datapoints are still highly overlapping in several plots. Also, I suggest that the authors consider including confidence intervals in the plots when possible.

Response: Following the journal's editorial guidance, we have provided source data for Fig. 2 and Fig. 4 with overlapping data points.

Reviewer #2 (Remarks to the Author):

In my view, the authors did a very thorough job of addressing the concerns the reviewers had with the initially submitted manuscript. The key finding of greater forest productivity for sites with mixed AM-ECM tree species was well-supported and potentially confounding factors were appropriately considered. The revisions greatly strengthened the introduction, the

interpretation of the results and the conclusions, as well as the overall flow of the manuscript. The resulting revised manuscript draws well-reasoned conclusions from a large dataset, with interesting implications for forest ecologists and managers.

Response: We thank the reviewer for this positive assessment!